# Machine Learning Models and Applications for Early Detection

**DOI:** 10.3390/s24144678

**Published:** 2024-07-18

**Authors:** Orlando Zapata-Cortes, Martin Darío Arango-Serna, Julian Andres Zapata-Cortes, Jaime Alonso Restrepo-Carmona

**Affiliations:** 1Instituto Tecnológico Metropolitano, Medellín 050034, Colombia; orlandozapata@itm.edu.co; 2Facultad de Minas, Universidad Nacional de Colombia, Medellín 050034, Colombia; mdarango@unal.edu.co (M.D.A.-S.); jarestrepoca@unal.edu.co (J.A.R.-C.); 3Fundación Universitaria CEIPA, Sabaneta 055450, Colombia

**Keywords:** machine learning models, early detection, data analysis, fraud detection, performance metrics

## Abstract

From the various perspectives of machine learning (ML) and the multiple models used in this discipline, there is an approach aimed at training models for the early detection (ED) of anomalies. The early detection of anomalies is crucial in multiple areas of knowledge since identifying and classifying them allows for early decision making and provides a better response to mitigate the negative effects caused by late detection in any system. This article presents a literature review to examine which machine learning models (MLMs) operate with a focus on ED in a multidisciplinary manner and, specifically, how these models work in the field of fraud detection. A variety of models were found, including Logistic Regression (LR), Support Vector Machines (SVMs), decision trees (DTs), Random Forests (RFs), naive Bayesian classifier (NB), K-Nearest Neighbors (KNNs), artificial neural networks (ANNs), and Extreme Gradient Boosting (XGB), among others. It was identified that MLMs operate as isolated models, categorized in this article as Single Base Models (SBMs) and Stacking Ensemble Models (SEMs). It was identified that MLMs for ED in multiple areas under SBMs’ and SEMs’ implementation achieved accuracies greater than 80% and 90%, respectively. In fraud detection, accuracies greater than 90% were reported by the authors. The article concludes that MLMs for ED in multiple applications, including fraud, offer a viable way to identify and classify anomalies robustly, with a high degree of accuracy and precision. MLMs for ED in fraud are useful as they can quickly process large amounts of data to detect and classify suspicious transactions or activities, helping to prevent financial losses.

## 1. Introduction

Machine learning (ML) has become a discipline that automates repetitive and complex tasks through its algorithms, thereby increasing operational efficiency across various organizations. ML analyzes large amounts of data to identify patterns and trends, aiming to improve decision making in different contexts [1,2,3,4,5,6,7]. This discipline of artificial intelligence trains models based on data analysis to make automatic predictions, allowing the models to deduce the correct labels based on the learning acquired from historical data.

Although ML has advantages over classical methods used in multiple areas, each ML model is unique. Each model is trained with data of different characteristics that must be identified to make correct predictions with a high degree of accuracy and precision (e.g., numerical, alphanumeric, and discrete data). Additionally, ML training takes a long time due to the large volumes of data that the models require [3].

Another associated disadvantage is the interpretability of the data [8], which limits the understanding of the model to make high-quality predictions. There is also an ongoing need for high-quality data, which must be sufficiently reliable and that effectively allows expression of the problem statement. In addition to this, the need to have a sufficient amount of data to train the models can be costly and time-consuming to collect [9]. There is also the risk of bias. Poor data quality during the training, validation, and testing can lead the model to make inaccurate decisions [10].

In general, despite the multiple disadvantages mentioned earlier, ML continues to have advantages in automating processes, improving decision making, and personalizing services by analyzing user preferences to enhance customer experience. ML can revolutionize businesses and society in general. Currently, ML is a trending tool that every organization must constantly monitor to remain relevant and increasingly competitive [11].

Early detection (ED) is an important field in ML. The implementation of accurate predictive models for early detection (ED) improves outcomes by optimizing time and resources, enhancing problem prevention, and contributing to the improvement of policies for early decision making [9]. ML-based early detection (ED) is currently actively used in socially impactful areas such as production processes, agronomy, energy efficiency, fraud detection, and medicine.

ML-based early detection (ED) in production processes can improve aspects such as quick intervention in automated processes, limiting negative impacts on operations, and preventing potential damages. It also allows cost savings by reducing downtime and cutting maintenance expenses. Additionally, it enables greater reliability in automated processes, ensuring consistent and accurate results, improving yields, and maximizing productivity and production [4,9].

ML-based early detection (ED) in agronomic processes is used to improve crop health, maximize resource management, increase yields, mitigate climate change, and promote sustainable agricultural practices. Farmers can ensure food security and environmental sustainability by proactively addressing challenges and making better decisions through the use of ML and new technologies [6,12].

In energy systems, ML-based early detection (ED) helps with optimizing energy consumption, reducing costs, and improving sustainability in various sectors such as electric grids and microgrids. ML algorithms are used to detect faults or anomalies at an early stage, aiding in fault prevention, improving reliability, and prolonging the lifespan of critical assets, such as power transformers [13]. Additionally, ED through ML can enable the implementation of preventive and proactive maintenance strategies, leading to greater operational efficiency and minimizing downtime in industrial energy systems [3].

ML-based early detection (ED) in medicine is used in public health programs to reduce healthcare costs and improve procedures with the aim of achieving more satisfactory outcomes in patients [14,15,16]. ML-based early detection (ED) has shown promising results in the prediction and early identification of diseases such as diabetes [17,18,19,20], cardiovascular diseases [21], breast cancer [10,18,22,23,24], and dementia [25]. 

Currently, medical personnel can detect diseases in the early stages, facilitating rapid interventions and the application of preventive measures more tailored to the symptoms of the diseases [26]. Additionally, improvements have been found in patient care, optimizing treatment plans [27], and achieving more accurate diagnoses. Continuously, ML promotes proactive healthcare management [15], which improves the sustainability of healthcare systems and contributes to health improvement initiatives.

Moreover, ML-based early detection (ED) in fraud detection is used for several reasons. It allows timely intervention, preventing further fraudulent activities and minimizing potential financial losses and damages [28]. It helps reduce negative impacts on organizations and financial systems by detecting fraudulent activities before they escalate into larger issues. Additionally, it allows for the saving of resources by preventing larger losses that may occur if fraud is detected at a later stage [29]. ML-based fraud ED helps maintain the integrity of data and financial records by restricting unauthorized access and manipulation of information [30]. ML-based ED provides valuable information about fraudulent patterns and trends to enhance fraud detection strategies. In general, ML-based ED allows for risk mitigation, asset protection, maintaining trust between entities, and ensuring security and stability in financial organizations and government fiscal surveillance systems.

Given the importance of ML-based ED, this article aims to conduct a systematic literature review to identify the most commonly used ML models (MLMs) in ML-based ED in the aforementioned areas. As a second objective, this article aims to identify new methodologies for using these MLMs to improve classification or prediction in ED. This literature review is motivated by the exploration of MLMs currently used in ED for fraud detection. The intention of this article is to identify how ML-based ED models have impacted the field of fraud detection by analyzing their advantages and disadvantages, and to present a discussion on the benefits that the fraud domain can obtain as part of Fiscal Surveillance and Control.

This review aims to provide academics and professionals with guidance in their work, facilitating the quick identification of current algorithms and methodologies used in ML for the application of ED in the referenced areas and in the field of fraud. The main contributions of this article are as follows:The presentation of the most used MLMs for ED.The division of MLMs for ED into two main categories: Single Base Model (SBM) and Stacking Ensemble Model (SEM).The identification of SBM or SEM in ED for fraud.A discussion on how ML-based ED can improve processes in fraud.

The article is structured in the following sections: Section 2 describes the research article selection process for a systematic literature review on MLMs for ED. Section 3 gives an overview of data balancing and model validation metrics currently used in machine learning. Section 4 gives an overview of the machine learning models found and their performance in multiple applications and specifically in fraud detection. Section 5 discusses the performance of the machine learning models found for early detection in multiple areas and the importance of using these models in early fraud detection. Finally, conclusions are presented.

## 2. Article Selection Process

A systematic literature review is a research approach that examines information and findings regarding a research topic [31]. This approach aims to locate the largest possible number of relevant studies on the subject of study and, through referenced or proprietary methodologies, determine what can be confidently asserted from these studies [32,33]. This section provides an overview of the literature to help understand the MLMs for ED used in the present literature. 

In this article, the process of searching and selecting articles consists of two stages, aiming to answer the following two questions:RQ1: Which MLMs are currently used in the literature for early detection in multiple areas?RQ2: How have these MLMs for ED been implemented in the context of fraud?

In stage 1, based on RQ1 and RQ2, Scopus was selected as the search engine, and the following keywords were used for the search equation: “machine learning model”, “data analytics” or “data analysis”, “early detection”, and, finally, “fraud detection”. These words were searched in the titles, abstracts, and keywords of the articles according to the use of the Scopus search engine. These five keywords formed the first search equation within the time frame between 2017 and 2025. As a result of this search, a total of 61 scientific articles were obtained, included in electronic databases such as Springer Link, Elsevier, IEEE, MDPI, and Taylor and Francis. 

Figure 1 presents an analysis of the occurrence of words in the selected articles. The cooccurrence analysis of the words of the articles was performed according to the information that was exported from the Scopus database. Information such as citation information: “Author(s), Document title, Year, Source title, etc.”, bibliographical information: “Abbreviated source title, Affiliations, etc.”, abstract and keywords: “Abstract, Author keywords and Indexed keywords”, funding details: “ Acronym, etc.”, and other information like “Conference information and Include references”. The color map in Figure 1 illustrates the frequency of recurring concepts found in the literature. It is inferred, then, that models such as Logistic Regression (LR), Support Vector Machines (SVMs), decision trees (DTs), data analytics (DA) and Random Forests (RFs) are base models used in ED or early diagnosis. 

Having reviewed the information from this preliminary search, in stage 2, it was identified that currently in ML, from a methodological aspect, with the aim of improving the prediction of base models, stacking ensemble has been implemented in recent years. This allowed extending the initial search equation by including the keyword “Stacking Ensemble” with a focus on ED. As a result, a total of 62 articles related to ML-based ED were obtained. Additionally, a search matrix was synthesized to identify the contribution of each article, the base MLMs (49 articles) and the ensemble MLMs (12 articles), and their application areas and one article of literature associated to the topic. Adding “Stacking Ensemble” provided only one more article, further completing the search for models in assembly. Once the review was advanced, the above keywords included multiple articles of the assembly models in an accurate manner, ensuring that the information on models under this methodology was mapped in the best possible way.

Finally, the necessary information was extracted regarding applications and MLMs applied to ED in multiple areas (49 articles) as shown in SBM and SEM tables, particularly in fraud detection Table (12 articles). 

## 3. MLM Data Balancing and Performance Metrics

In the literature review consulted, various procedures were evident for conducting proper validations of MLMs, such as data balancing and the application of performance metrics. Validating MLMs is crucial to ensure their reliability and effectiveness in decision making. Validation allows for evaluating the predictive capacity of models, identifying potential issues like overfitting, and ensuring that the results obtained are generalizable and consistent with new data [34]. In model validation, its performance, accuracy, and capability to handle different scenarios can be verified, which helps ensure that it is useful and reliable in real-world applications [17].

### 3.1. Data Balancing

Data balancing in MLMs addresses the problem of class imbalance. One class may have significantly more instances than another in a dataset. When classes are imbalanced, models tend to favor the class with more instances, which can lead to poor performance in predicting the class with fewer instances.

Using data balancing techniques in the preprocessing of information to train an ML model improves the model’s ability to learn patterns from all classes equitably, resulting in more accurate, precise, and generalizable predictions. The purpose of balancing is to achieve an equilibrium where the detection of both minority and majority classes is of interest.

Data balancing can be achieved through techniques such as oversampling (duplicating instances of the minority class), undersampling (removing instances of the majority class), or more advanced methods like SMOTE (Synthetic Minority Over-sampling Technique) [35]. These techniques help improve the predictive capability of models by ensuring that all classes are treated equitably during training.

### 3.2. Performance Metrics

Performance metrics in ML are measures used to evaluate the performance and effectiveness of machine learning models in data prediction and classification. In the literature review consulted, metrics such as accuracy, precision, recall (sensitivity), specificity, F1-score, AU-ROC (area under receiver operating characteristics curve), AU-PRC (area under the precision–recall curve), the MCC (Matthews correlation coefficient), and the confusion matrix [34] are used. These metrics provide information about the predictive capability, accuracy, and overall effectiveness of MLMs. All the mentioned metrics are related to the confusion matrix. The confusion matrix (Table 1) is a tool that allows visualizing the performance of a classification model by showing the number of true positives (TP), true negatives (TN), false positives (FP), and false negatives (FN) that the model has produced on a test dataset [22]. Table 1 presents the confusion matrix for a binary problem.

The confusion matrix can also be extended to multiclass problems. It is not necessarily intended only for binary problems. In a multiclass classification context, the confusion matrix is expanded to include all classes present in the problem. Thus, its construction will have a length of *N*, corresponding to the number of classes. Table 2 presents the descriptions of the evaluation metrics.

## 4. Machine Learning Models for ED and Applications

According to the literature review, two methodologies for applying MLMs in multiple applications were identified. These methodologies describe the use of MLMs from the following perspectives.

Single Base Models (SBMs). These base models serve as individual classifiers or regressors that make predictions based on input data. Among the most common base models are LR (Logistic Regression), SVM (Support Vector Machine), DT (decision tree), RF (Random Forest), NB (naive Bayes), K-Nearest Neighbors (KNNs), and neural networks (NNs). When using a single model, the choice depends on the characteristics, distribution, and properties of the datasets [16]. SBMs are also used as a basis for ensemble methods and more complex stacking models [36]. Table 3 presents a brief description of SBMs.

Stacking Ensemble Models (SEMs). They involve combining multiple base models to improve the predictive performance of SBMs. These models use a two-level stacking approach; base models make predictions at the first level, and meta-learning combines these predictions at the second level [13,16]. The purpose of this ensemble methodology is to combine two or more models, each with its strengths and weaknesses, to construct a more robust model. Stacking ensemble models have proven to be promising in various applications by offering advantages such as improved accuracy, reduced overfitting, and enhanced performance compared to individual models [37]. SEMs use boosting, bagging, and stacking schemes [30]. Each SEM operates within its own domain space, showing varying levels of performance based on the aggregated selection of base models and the distribution, nonlinearity, and class imbalance present in the dataset. Some of the most popular boosting algorithms are AdaBoost (Adaptive Boosting), Gradient Boosting, XGBoost 2.0.1, and LightGBM 4.4.0 [13]. These algorithms have variations in how they adjust weights and combine weak models to form the final model, but they follow the general scheme of boosting. An example of bagging is RF, where multiple decision trees are trained on training datasets generated by bootstrap sampling (sampling with replacement), and predictions from individual trees are averaged to produce the final prediction.

**Table 3 sensors-24-04678-t003:** Description of SBM and XGBoost of SEM.

Model	Description
LR [38]	A statistical model used to analyze the relationship between a dependent variable (binary outcome) and one or more independent variables. It is commonly used for binary classification tasks where the outcome variable is categorical with two possible outcomes. Logistic regression estimates the probability that a given input belongs to a specific category by fitting the data to a logistic function, which transforms the outcome into an interval between 0 and 1.
SVM [39]	A supervised machine learning algorithm used for classification and regression tasks. SVM works by finding the optimal hyperplane that best separates data points into different classes in a high-dimensional space. Its goal is to maximize the margin between the classes, making it effective for both linear and nonlinear classification problems. SVM can handle high-dimensional data and is known for its ability to generalize well to unseen new data.
DT [40]	A machine learning algorithm used for classification and regression tasks. It is a tree-shaped model where internal nodes represent features, branches represent decisions based on those features, and leaf nodes represent the outcome or decision. The algorithm recursively splits the data based on the most significant feature at each node, aiming to create homogeneous subsets. Decision trees are easy to interpret and visualize, making them valuable for understanding the decision-making process in a model. They can handle both numerical and categorical data, making them versatile for various types of datasets.
RF [41]	A machine learning algorithm composed of multiple decision trees. Each tree is built using bootstrapping and random feature selection to create an ensemble of uncorrelated trees, resulting in more accurate predictions than individual trees. The algorithm leverages the concept of collective knowledge, where the forest of decision trees works together to make predictions, and the final prediction is based on the majority vote of the trees.
NB [42]	A probabilistic classifier based on the application of Bayes’ theorem. It assumes that the presence of a particular feature in a class is not related to the presence of any other feature. Despite their simplicity, naive Bayes classifiers are known for their efficiency and effectiveness in various classification tasks, especially in text classification and spam filtering.
KNN [43]	A machine learning algorithm used for classification and regression tasks. In KNN, the class or value of a data point is determined by the majority class or the mean value of its nearest neighbors in the feature space. The algorithm calculates the distance between data points and classifies them based on the majority class of the nearest k data points.
ANN [44,45]	A computational model inspired by the structure and functioning of the neural networks in the human brain. ANNs consist of interconnected nodes, known as artificial neurons, that process information and learn patterns from data. These networks are used in machine learning and deep learning to solve complex problems such as pattern recognition, classification, and regression, among others.
XGBoost [46,47]	A model that uses gradient boosting to optimize the loss function and handle complex patterns in data. XGBoost is widely used for classification, regression, and ranking tasks due to its speed, accuracy, and ability to handle large datasets efficiently. It uses decision trees as base models and trains them sequentially. XGBoost in some cases is considered a base model grounded in DT.

Within the literature review, different types of SBMs and SEMs were found. Under the search parameters in the matrix of sintering literature review, for ED, different configurations of these models were obtained in areas such as medicine (37 articles), fraud detection (11 articles), agronomy (2), energy efficiency (2 articles), industrial processes (2 articles), education (1 article), and telecommunications (1 article). Table 4 presents Multidisciplinary MLM -SBM for early detection, for multiple areas excluding articles related to fraud detection, which are analyzed later. Table 5 presents Multidisciplinary MLM-SEM for early detection.

According to Table 4, the most used SBMs for early detection, based on the reviewed literature, are Random Forest (RF), Support Vector Machine (SVM), and K-Nearest Neighbor (KNN). These models are applied in the medical field, achieving average performances above 80% according to the metrics reported by various authors. In other fields, RF also stands out as a frequently used model, with performance levels similarly approaching 80%.

On the other hand, in the SEMs presented in Table 5, it is identified that RF is the most commonly used base model, followed by DT, then LR, SVM, and KNN with the same frequency, followed by ANN and XGB also at the same level. NB is the least used base model for early detection in the literature studied. As the best meta learner, LR is identified as the most used due to its simplicity in computation and inference for decision making. It is also identified that the SEM methodology achieves performances in some cases exceeding 90%.

According to Table 5, the SEM yields superior results compared with the SBM. It is important to note that the SEM approaches found are implemented in the medical field, which is a critical area for decision making, as a misdiagnosis can have severe repercussions in various health contexts. This justifies why most of the literature on SBMs and SEMs consulted focuses on the field of medicine, in comparison with other areas for early detection.

Hybrid models acting as MLMs and SEMs were also found. The selection of base, ensemble, or hybrid models will depend on the working context and the characteristics of the data.

### Machine Learning Models for ED in Fraud Detection

MLMs are used in fraud detection to analyze patterns and behaviors in data in order to identify fraudulent activities. These models are important for early fraud detection (ED) as they can quickly process large amounts of data to detect and classify suspicious transactions or activities, helping to prevent financial losses.

ML-based models for ED offer several advantages over traditional fraud detection methods. MLMs have the ability to adapt and improve over time as more data are analyzed, thereby increasing their accuracy in fraud detection [69]. They can also analyze complex and diverse data sources, allowing them to detect sophisticated and evolving fraud schemes that may go unnoticed by traditional rule-based systems or human intervention [70]. Table 6 presents the MLMs found in the literature for ED for fraud detection.

Table 6 presents the MLMs found in the literature consulted for ED in fraud detection. Only the use of SBMs was found for fraud detection. RF persists as the most frequently used model, followed by KNN. LR, SVM, and XGB are used at the same level for fraud detection. RF continues to be the most used and reliable model for fraud detection according to Table 5, with two authors reporting better performance with this model. Although RF is based on bagging and XGB utilizes boosting, they are considered base models grounded in DT. Table 6 reflects that in most cases, each MLM achieved a high number of accurate fraud detections in real fraud cases, with authors reporting accuracy metrics exceeding 90%.

All the MLMs found in Table 4, Table 5 and Table 6 were evaluated using the metrics reported in Table 2. Each achieved positive performance metrics in the respective contexts where they were implemented.

## 5. Discussion

ML algorithms are increasingly used in various fields due to their ability to adapt to new data and identify hidden patterns, enabling decision making with a higher degree of reliability. Although most models found in the literature work as standalone base models, the use and experimentation with ensemble methodologies to improve the performance of base models is becoming increasingly common.

The information in Table 4 and Table 5 shows that base models are used diversely across multiple areas, unlike SEMs. Specifically, Table 5 reports that SEMs work exclusively in the medical field. This is justified because SEMs, by gathering the various decisions from SBMs, allow for a more robust acquisition of data variability and a better fit to the data. Consequently, the identification of the problem is more accurate and precise, whether in the context of regression or classification. The unification of these decisions constitutes a more solid knowledge base that serves as input to another model for final decision making. This is particularly important in the medical field, where decision making is critical for diagnosing a person, requiring a minimal margin of error.

Within the models in Table 6 for early fraud detection, it was found that the models found in the literature are SBMs. Although the authors do not consider SEMs for early fraud detection, the SBMs used achieved significant performance with good adaptability to emerging patterns, good training times, and good adaptation to data for fraud detection [1,70,71,73,74,75].

An important aspect to achieve satisfactory results in the training, validation, and testing of SEMs and SBMs in any context will be the associated data engineering analysis [77]. This refers to the effective selection of the data characteristics that would be supplied as information to the SBM. Additionally, analysis of data balancing techniques is used to balance major or minor classes [69], such as the SMOTE method [74]. 

This aspect of SEMs is not considered critical since the patterns of training, validation, and testing are responses from the SBM. However, in this case, cross-validation processes must be ensured to avoid overfitting issues, model selection bias, and errors in variance estimation. The use of SEMs must strictly consider cross-validation methods such as K-Fold, Hold-Out, Leave-One-Out, Leave-P-Out, Monte Carlo, Stratified K-fold, and Repeated K-fold, Time Series Cross-Validation, and Nested Cross-Validation [25,52]. These methods provide a more reliable estimation of model performance on unseen data, reducing the risk of overfitting and enabling better hyperparameter tuning and model evaluation.

The advantages of machine learning models in early anomaly detection include their ability to process large amounts of data quickly, identify suspicious patterns, and adapt and improve over time.

However, some disadvantages may include the lack of labeled data available and the complexity in identifying fraudulent operations. A brief description of some advantages, disadvantages, and areas for improvement for the models found in the literature for early anomaly detection is presented below.

*Logistic Regression.* **Advantages:** Simple and easy to interpret. It is efficient in detecting linear patterns in data and can be useful in situations where relationships are simpler and more direct. **Disadvantages:** Not effective in detecting anomalies in highly imbalanced datasets or datasets with complex characteristics. **Improvement directions:** Combining with other supervised or unsupervised learning techniques can improve its performance, especially in situations where relationships are more complex.*Support Vector Machine.* **Advantages:** It is effective in identifying complex and nonlinear patterns in data. Additionally, it can handle high-dimensional datasets. **Disadvantages:** Requires longer training time compared to other algorithms and can be sensitive to the choice of hyperparameters, affecting its performance. **Improvement directions:** Parameter optimization techniques and data balancing can improve the model’s performance.*Decision tree.* **Advantages:** Easy to interpret and visualize, which facilitates understanding of how decisions are made in anomaly detection. They can effectively handle mixed data, including categorical and numerical data. **Disadvantages:** They tend to overfit and exhibit high sensitivity to small changes in input data. **Improvement directions:** Incorporating regularization techniques can mitigate overfitting and improve generalization in anomaly detection. Combining multiple decision trees into an ensemble, such as Random Forest or Gradient Boosting, can enhance model accuracy and robustness.*Random Forest.* **Advantages:** It can effectively handle imbalanced datasets and reduces the tendency to overfit. **Disadvantages:** The complexity of Random Forest can make it difficult to interpret how decisions are made and sometimes requires careful hyperparameter tuning. **Improvement directions:** Perform comprehensive hyperparameter optimization to improve the model’s detection capability and combine it in an ensemble with other models to significantly enhance decision making.*Naive Bayes.* **Advantages:** Computationally efficient, can handle large datasets, and can effectively manage high-dimensional datasets. **Disadvantages:** Highly sensitive to noisy or outlier data. **Improvement directions:** Use more advanced versions of naive Bayes, such as Kernel naive Bayes or Multinomial naive Bayes, which can improve detection capability.*K-nearest neighbor.* **Advantages**: Simple, easy to implement, and can identify nonlinear patterns in the data. **Disadvantages**: Sensitive to noisy and outlier data. **Improvement directions**: It can benefit from parameter optimization, such as the number of neighbors. Using weighting methods to give more weight to closer neighbors can reduce the impact of outlier data in anomaly detection.*Artificial neural network.* **Advantages**: This model can learn complex and nonlinear patterns in the data, and can effectively handle structured and unstructured data. **Disadvantages**: They require large amounts of data, and the complexity and training time of ANNs can be significant. **Improvement directions**: Implement regularization techniques and hyperparameter optimization to improve generalization and performance. Exploring ensemble learning approaches that combine multiple neural networks can enhance the model’s accuracy and robustness.*XGBoost.* **Advantages:** High performance and accuracy, effective handling of imbalanced datasets. **Disadvantages:** May require careful hyperparameter tuning, and the model’s complexity and training time can be significant. **Improvement directions:** Combining with other models in an ensemble can lead to significant improvements in anomaly detection.*Stacking Ensemble.* **Advantages:** Combines multiple models to improve accuracy in early anomaly detection. The combination of models allows for greater robustness and generalization in anomaly detection. **Disadvantages:** Implementation can be more complex than a single model, and there is a risk of overfitting when combining multiple models. **Improvement directions:** Perform comprehensive optimization of model combinations in Stacking Ensemble to enhance early anomaly detection, and integrate regularization techniques to mitigate the risk of overfitting.

The use of MLMs in ED for fraud detection, whether as SBMs or SEMs, will continue to be the subject of study in areas such as corporate security, surveillance, and fiscal and financial control, due to their ability to process large amounts of data rapidly and adapt to new information over time.

A determining factor in MLMs’ training in fraud ED is the limitations in data collection for model training. Data collection for fraud ED is constrained by the availability of labeled data due to the lack of digitalization of this information. Without digitized data, the process of consolidating a labeled historical dataset is slow and costly, which restricts the applicability of MLMs in fraud ED, especially in cases such as tax fraud detection [29]. The lack of digitized data can hinder the effectiveness and accuracy of fraud ED processes, as manual data handling is time-consuming and error-prone. Moreover, the challenge of labeling transactions as fraudulent or non-fraudulent can be complex due to the difficulty in definitively asserting the fraudulent nature of transactions, leading to careful use of labeled examples and the need for verification by expert personnel to identify such specific fraud, which is also subject to ethical considerations [69].

It is important that in the field of fraud detection in ED, strategies for data balancing will be considered based on available information to reduce intrinsic bias that may include human manipulation or unwanted value judgments when labeling data.

Implementing SEMs as MLMs for fraud ED offers several advantages over base models and traditional methods. Some of the advantages that this type of model can offer include:Improved prediction performance by maximizing fraud detection through effective identification of patterns and anomalies in the data, leading to better prediction performance. Using features based on SBM responses allows the analysis of behaviors that may not have been explored in traditional methods, thereby enhancing a more comprehensive analysis of fraud indicators. Additionally, the adaptability and robustness of SEMs enable them to adjust to the strengths and weaknesses of multiple baseline models, improving overall detection performance and robustness in identifying fraudulent activities [16].Combining the predictive power of various models enables the identification of fraudulent behaviors at an early stage with greater accuracy, which allows for timely intervention and prevention of fraudulent activities [20].SEMs provide a more reliable balance between precision and interpretability, making them operationally viable for fraud detection tasks, due to the adoption of the features and operating dynamics of SBMs.

## 6. Conclusions

The literature review under the search equation allowed for the consolidation of two ways of using MLMs for ED: Single Base Models (SBMs) and Stacking Ensemble Models (SEMs). The implementation of SBMs was identified in different areas, whereas SEMs were implemented only in the field of medicine due to the high precision required in this area.

The implementation of SEMs can favor and strengthen conventional fraud detection efficiently, allowing to improve prediction performance by leveraging the integration of features from different base models. An SEM enables better adaptability to data and robust decision making. Additionally, it provides more accurate early detections in scenarios with high data variability, reduces issues such as overfitting, and allowing handling biases in the data.

Both SBMs and SEMs have proven to be efficient in early detection across multiple areas, particularly in fraud, with accuracies in some cases exceeding 90%. For the use of MLMs, it will always be relevant to perform data engineering processes to select appropriate features for model training, and to pay special attention to data balancing to achieve adequate results in predictions.

From the analyzed information, it can be inferred that a challenging task in the field of fraud detection is the consolidation of reliable databases for training MLMs for ED, as well as the adoption of new models and cutting-edge methodologies such as deep learning.

Future research lines may be oriented to develop further advancements in techniques and technologies like deep learning models as SBMs and SEMs to enhance the accuracy, efficiency, and scalability of fraud detection systems. Other potential research lines include: Enhanced Data Enrichment (the quality and quantity of data), Advanced Machine Learning Algorithms in fraud detection (convolutional neural networks and ANNs, Long Short-Term Memory), Real-time Fraud, and Blockchain Technology for secure and transparent transaction verification and enhancing fraud detection capabilities through immutable and decentralized data storage.

Another future line of research is to focus the analysis of machine learning models according to specific areas of application. As found, there are a higher number of studies available in the field of medicine compared to any other area. In this way, being specific in the search can provide a selective scope that can be important for specific researchers. In addition, it will be important as future work to consider other considerations to obtain a greater number of articles from areas different than medicine, to obtain a more comparable analysis in these types of models and methodologies.

## Figures and Tables

**Figure 1 sensors-24-04678-f001:**
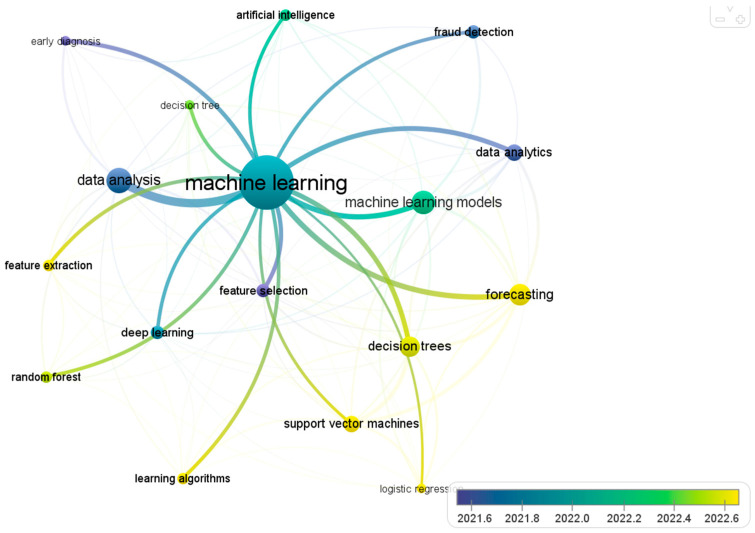
Analysis of the occurrence of words in the literature review.

**Table 1 sensors-24-04678-t001:** Confusion matrix for a binary problem.

	Positive Class (Prediction)	Negative Class (Prediction)
Positive class (Outcome)	TP	FN
Negative class (Outcome)	FP	TN

**Table 2 sensors-24-04678-t002:** Descriptions of the evaluation metrics.

Metric	Description	Formulation
Accuracy	Proportion of correct predictions out of the total predictions made by the model.	$ACC=\frac{TP+TN}{TP+TN+FP+FN}$
Precision	Proportion of true positives (TP) over the sum of true positives and false positives (FP).	$P=\frac{TP}{TP+FP}$
Recall (Sensitivity)	Proportion of true positives to the sum of true positives and false negatives (FN).	$R=\frac{TP}{TP+FN}$
Specificity	Proportion of true negatives (TN) over the sum of true negatives and false positives (FP).	$S=\frac{TN}{TN+FN}$
F1-Score	It is the harmonic mean of precision and recall.	$F1=2\cdot \frac{P.R}{P+R}$
AU-ROC	ROC chart represents the true positive rate (TPR) versus the false positive rate (FPR) at various thresholds. A higher AU-ROC indicates better model performance.	(TPR) Vs (FPR)
AU-PRC	PRC chart shows the relationship between precision (P) and recall (R) for different classification thresholds of the model. A higher AU-PRC indicates better model performance.	(P) Vs (R)
MCC	Correlation between true classes and predicted labels.	$MCC=\frac{TP.TN-FP.FN}{\sqrt{\left(TP+FP\right).\left(TP+FN\right).\left(TN+FP\right).(TN+FN)}}$

**Table 4 sensors-24-04678-t004:** Multidisciplinary MLM—SBM for early detection.

Ref.	Application	Dataset	Best Model (*)-Other Models (+)									Evaluation Metric Best Model	Area
			LR	SVM	DT	RF	NB	KNN	ANN	XGB	+		
[48]	Alzheimer’s disease detection	Kaggle Alzheimer’s disease prediction dataset includes age, gender, status of income, MMSE, eTIV		x		x			* x			ACC:88% P:NA R:NA F1:NA AUCROC:NA	Medicine
[49]	Forecasting coronavirus	COVID dataset includes date, time, state/union territory, confirmed Indian national, cured, deaths, and confirmed			x			x	* x		^1^ PR	RMSE:78 ACC:NA P:NA R:NA F1:NA AUCROC:NA	
[50]	Prognostic prediction of Alzheimer’s disease	Alzheimer’s Disease Neuroimaging Initiative (ADNI) database—data of 806 participants		x				x	* x			ACC:53.5 P:NA R:NA F1:NA AUCROC:NA	
[51]	Predict the early onset of heart failure	Retrospective EHR dataset where all patients (4370 cases and 30,132 controls) and all data domains were included in these experiments—256 features	x			x			* x			ACC:NA P:NA R:NA F1:NA AUC:77	
[18]	Breast cancer, heart disease, and diabetes detection	Diabetes dataset from Kaggle, ensuring that it includes all relevant features such as age, sex, body mass index (BMI), blood pressure, serum insulin level, and glucose level	x	x	* x			x				ACC > 90 P:NA R:NA F1:NA AUCROC:NA	
[52]	Intrauterine Fetal Demise detection	Cardiotocography dataset		x	x	x	x	x	x		* ^2^ GB	ACC:99 P:99 R:98 F1:99 AUCROC:NA	
[34]	Predicting Abnormal Respiratory Patterns in Older Adults	Abnormal breathing patterns in older adults dataset—25,000 records, 3 features		x				x			* ^2^ GB	ACC:100 P:100 R:99 F1:99 AUCROC:100	
[17]	Detection and Accurate Classification of Type 2 Diabetes	Kaggle-hosted Pima Indian dataset. Eight features, 769 rows	x	x	x		x	* x				ACC:72 P:57 R:79 F1:67 AUCROC:NA	
[26]	Classification-based screening of depressive disorder patients through graph, handwriting, and voice signals	Handwriting and voice tasks			x			* x				ACC:78 P:NA R:NA F1:NA AUCROC:NA	
[53]	Glaucoma recognition	Glaucoma diagnosis dataset based on vital 45 features of OCT images from a combination of public and private datasets		x		x		* x		x		ACC:99 P:89 R:1 F1:94 AUCROC:NA	
[22]	Early detection of breast cancer	Breast cancer wisconsin dataset was collected from Kaggle repository. The dataset contains 32 features and 600 instances	* x	x	x	x	x	x				ACC:94 P:93 R:97 F1:95 AUCROC:97	
[54]	Malnutrition Risk Assessment in Frail Older Adults	Mini Nutritional Assessment SF dataset	* x	x		x		x			^3^ AB	ACC: 90 P:83 R:86 F1:NA AUCROC:NA	
[55]	Polycystic Ovary Syndrome Prediction	Polycystic ovary syndrome dataset—541 patients, 41 features	x	x		x		x	x		* ^4^ GNB,^2,5^	ACC:100 P:100 R:100 F1:100 AUCROC:80	
[25]	Early detection of dementia	Alzheimer’s dataset—9 features, 12 rows.			x		* x	x	x			ACC:100 P:NA R:NA F1:NA AUCROC:NA	
[14]	Early detection of complications after pediatric appendectomy	Appendicitis dataset	x	x		* x	x	x		x	^2,3,5^	ACC:83 P:NA R:NA F1:NA AUCROC:80	
[56]	Early detection of hepatocellular carcinoma and cirrhosis	Volatile organic compounds (VOCs) dataset				* x						ACC:85 P:NA R:80 F1:NA AUCROC:0.96	
[21]	Predicting heart disease	Dataset of patient medical records that included blood pressure, cholesterol levels, and other risk factors	x	x		* x				x		ACC:94.15 P:92 R:94 F1:93 AUCROC:0.96	
[57]	Diagnostic classification of hepatitis C tests	HCV dataset—dataset consists of hepatitis C test records of 615 patients. The patients’ records consisted of 238 women and 377 men with the age bracket of 19 to 77 years. The dataset contains 13 features	x		x	* x	x	x				ACC:98.9 ACC:85 P:NA R:NA F1:NA AUCROC:100	
[58]	Prediction of diseases and recommending drugs in healthcare	FDA Adverse Event Report System dataset, 6 features.	x		x	* x				x	^2^ ^,5^	ACC:85 P:97 R:96 F1:96 AUCROC:NA	
[2]	Prediction of heart attack	Heart attack dataset to predict whether a person can suffer from heart attack or not		* x	x			x				ACC:85 P:NA R:NA F1:NA AUCROC:NA	
[8]	Prediction of platinum resistance in ovarian cancer treatment	Demographic, clinicopathological, and laboratory findings of 102 patients with EOC, 46 features		* x		x						ACC:NA P:NA R:100 F1:NA AUCROC:99.3	
[59]	Prediction of post-stroke depression	Post-stroke depression dataset (development dataset: 775; test dataset: 194)								* x		ACC>81 P:NA R:NA F1:NA AUCROC:NA	
[60]	Early detection of heart failure	Heart failure with reduced ejection fraction dataset—90,357 adult noncardiac surgical procedures reviewed	x			x				* x		ACC:80.8 P:NA R:80.8 F1:NA AUCROC:87	
[27]	Early detection of cervical cancer	Cervical cancer dataset—36 features		x	x		x	x		* x	^3^	ACC:85 P:87 R:84 F1:NA AUCROC:91.2	
[61]	Stroke analysis and prediction in the healthcare industry	Stroke prediction dataset—4 features	x	x		* x				x		ACC:0.92 P:95 R:98 F1:96 AUCROC:91.2	
[62]	Breast cancer diagnosis	Wisconsin diagnostic breast cancer dataset—30 numerical features							* x		^2^	ACC:98.5 P:NA R:NA F1:NA AUCROC:NA	
[63]	Early Autism Spectrum Disorder prediction	Autism Spectrum Disorder (ASD) dataset—500 records, 10 features	* x	x	x	x			x	x	^3^	ACC:88 P:98 R:100 F1:99 AUCROC:NA	
[64]	Alzheimer’s early detection	Oasis longitudinal (4 features) and MRI image dataset Kaggle	x	x	x	x	x	x		* x	^3^	ACC:84 P:86 R:83 F1:84 AUCROC:NA	
[3]	Prediction of stochastic climate factors	Full Factorial Design microgrid dataset—3 features				* x						RMSE:7.029 ACC:NA P:NA R:NA F1:NA AUCROC:NA	Energy
[13]	Power Transformer Fault Detection	Transformer fault diagnosis, 12 features	x	x	x		x	x	x		* ^6^ GP	ACC>80 ACC:NA P:NA R:NA F1:NA AUCROC:NA	
[6]	Tail biting outbreak predictions in pigs using feeding behavior records	Data collected from 65 pens originating from two herds of grower-finisher pig		x		x		* x	x			ACC:96% P:NA R:NA F1:NA AUCROC:NA	Agronomy
[12]	Prediction of biological species invasion	Rainbow trout species invasion presence–absence dataset, 12 features	x		x	* x			x	x	^2^	ACC:NA P:NA R:NA F1:NA AUCROC:89	
[9]	Production line sampling prediction	Automotive industry manufacturer producing gearbox components, 4300 samples				* x						ACC:84 P:NA R:NA F1:NA AUCROC:NA	Industry
[4]	Predict failures in the production line.	Numeric sensors dataset, 3 features									* ^7^ IF	ACC:98 P:98 R:100 F1:98 AUCROC:NA	
[5]	Fraud Detection on Streaming Customer Behavior Data	Data collected from the City of Milan. These collected data include the collective usage data of subscribers belonging to that region from various regions									* ^8^ DS	ACC:99 P:NA R:NA F1:NA AUCROC:NA	Telecom
[65]	Early detection and mitigation of potential threats from near-Earth objects	Asteroid characteristics dataset—Kaggle dataset. A total of 958,525 rows and 45 features	x			x					* ^2^	ACC:99.9 P:100 R:100 F1:100 AUCROC:NA	astrogeology
[7]	Predict student academic performance	UC Irvine machine learning repository—1044 students’ academic performance in two high schools: demographic-, social-, and academic-related features		x	x	x			x	* x	^3,5^	ACC:97.12 P:NA R:NA F1:NA AUCROC:NA	academy
		FREQUENCY MODEL	16	21	16	22	9	19	12	11	NA: Not Available		
		FREQUENCY AS BEST MODEL	3	2	1	9	1	4	5	5			

^1^ Polynomial Regression; ^2^ Gradient Boosting; ^3^ AdaBoost; ^4^ Gaussian naive Bayes; ^5^ Stochastic Gradient Descent; ^6^ Gaussian process; ^7^ Insolation Forest; ^8^ DenseStream.

**Table 5 sensors-24-04678-t005:** Multidisciplinary MLM—SEM for early detection.

Ref.	Application	Dataset	Base Learners									Best Meta Learner	Evaluation Metric Best Model	Area
			LR	SVM	DT	RF	NB	KNN	ANN	XGB	+			
[66]	Detection of influenza disease	NIAID Influenza Research Database (IRD), 18,462 records	x	x		x		x	x		^9^	SVM	ACC:84.7 P:NA R:NA F1:83 AUCROC:83	Medicine
[10]	Early detection of breast cancer	Breast cancer dataset—569 instances, 10 real-valued features	x	x	x		x				^10^	^11^ DSS	ACC:96.2% P:NA R:96.3 F1:NA AUCROC:NA	
[23]	Classification of breast cancer	The Wisconsin Breast Cancer-Original (WBCO) data contained nine features and one diagnostic value for 699 breast biopsies				x		x			^2,12^ GLM	GLM	ACC:97.96 P:NA R:NA F1:NA AUCROC:NA	
[67]	Brain MRI analysis for Alzheimer’s disease diagnosis	OASIS dataset, 416 data samples							x			ANN	ACC:93 P:94 R:93 F1:92 AUCROC:NA	
[37]	Detection using images of the palpable palm	Palpable palm images (dataset) collected from 710 participants in selected hospitals in Ghana		x	x	x	x		x			NB	ACC:99.9 P:100 R:100 F1:100 AUCROC:NA	
[16]	Predicting emergency readmission of heart disease patients	Private dataset from the MIT Laboratory for Computational Physiology, not adopted in clinical studies on heart failure and cardiovascular disease, 13 features	x	x	x	x		x		x		XGB	ACC:88 P:NA R:74 F1:84 AUCROC:88	
[35]	Early detection of patients with alcohol use disorder	Alcohol use disorder (AUD) 2551 patients, 28 features		x	x	x		x		x		^13^ LIR	ACC:98 P:97 R:96 F1:97 AUCROC:99	
[19]	Diabetes prediction	Pima diabetes dataset 768 rows, 8 features								x	^3,13^ GBT	GBT	ACC:83.9 P:83.7 R:76.7 F1:78.3 AUCROC:87.7	
[36]	Heart abnormality detection	Heart attack dataset—11 features, 1190 rows	x		x	x			x		^15^ CB	LR	ACC:94 P:NA R:NA F1:NA AUCROC:92	
[68]	Cough Sound-based COVID-19 Detection	Coughvid dataset—25,000 cough recordings			x	x		x				LR	ACC:99.8 P:NA R:NA F1:99.8 AUCROC:99.8	
[20]	Early Diabetes Prediction	Diabetes dataset UCI’s machine repository—12 features				x					^3,14^	LR	ACC:97 P:NA R:96 F1:NA AUCROC:96	
[24]	Diagnosis of breast cancer	Clinical factors and BIRADS breast density—173,330 records, 14 attributes	x			x				x		LR	ACC:NA P:NA R:71.3 F1:NA AUCROC:76	
FREQUENCY MODEL			5	5	6	9	2	5	4	4	NA: Not Available			

^1^ Polynomial Regression; ^2^ Gradient Boosting; ^3^ AdaBoost; ^4^ Gaussian naive Bayes; ^5^ Stochastic Gradient Descent; ^6^ Gaussian process; ^7^ Insolation Forest; ^8^ DenseStream; ^9^ Boosting Trees; ^10^ Lasso Regression; ^11^ Decision Support System; ^12^ Generalized Linear Model; ^13^ Linear Regression; ^14^ Gradient Boosted Tree; ^15^ CatBoost.

**Table 6 sensors-24-04678-t006:** MLM for early detection in fraud.

Ref.	Application	Dataset	Best Model (*)-Other Models (+)									Evaluation Metric Best Model	Area
			LR	SVM	DT	RF	NB	KNN	ANN	XGB	+		
[28]	Fraud detection in utility companies	Customers’ consumption history: 2107 records, 14 features	x	* x		x	x	x	x	x	^16^ CART	ACC:62.3 P:NA R:72 F1:60.3 AUCROC:NA	Fraud
[71]	Enterprise Financial Audit	Audit risk dataset, 786 rows		x		* x			x			ACC:84.35 P:NA R:NA F1:NA AUCROC:NA	
[72]	Analysis of Fraud Detection in Healthcare	Inpatients, Outpatients, and Beneficiaries.	x		x	x				* x		ACC:87.5 P:42 R:89 F1:57 AUCROC:NA	
[73]	Credit Card Fraud Detection	The dataset holds transactions made by credit card holders in limited time, where 362 transactions are fraudulent out of 1136 transactions	x	x	* x			x				ACC:100 P:100 R:100 F1:100 AUCROC:NA	
[74]	Fraud detection problem in credit cards	IEEE-CIS fraud detection: 59,054 instances, 433 features and credit card fraud detection: 284,807 instances, 31 features.	x			x				x	* ^15^	ACC:NA P:81.2 R:94.1 F1:78.9 AUCROC:NA	
[1]	Big Data Analytics for Credit Card Fraud Detection	German credit card and Taiwan credit card datasets				x		* x			^2,3^	ACC:96.29% P:96.29 R:100 F1:NA AUCROC:NA	
[70]	Fraud in financial institutions	PaySimKaggle: 6,351,193 rows, 7 features		x		* x		x			^5^	ACC:98 P:97 R:90 F1:NA AUCROC:NA	
[69]	Fraud detection in financial and banking systems	Quinten’s partner about cheque fraud		x	x	x	x	x	x	* x	^13^	ACC:0.067 P:0.056 R:77.8 F1:NA AUCROC:NA	
[75]	Fraud Detection in Credit card and Transactions	Provided by Vesta Corporation, 11 features	x				x	x			* ^16^	ACC:96 P:96 R:96 F1:96 AUCROC:NA	
[29]	Detection of under-declarations in tax payments	A total of 1367 tax declarations of building projects in the city of Bogotá, Colombia—6 features									* ^17^ SC	ACC:NA P:NA R:NA F1:NA AUCROC:NA	
[76]	Fraud Detection in Digital Banking	Banking dataset from Kaggle-8 features									^2^	ACC:96 P:12 R:89 F1:21 AUCROC:NA	
[30]	Detection of earnings manipulation in financial firms	SEBI (Security and Exchange Board of India) reports and the LexisNexis database				x				x	* ^3^	ACC:60.5 P:NA R:60 F1:NA AUCROC: 74	
FREQUENCY MODEL			5	5	3	8	3	6	3	5			
FREQUENCY AS BEST MODEL			0	1	1	2	0	1	0	2			

^1^ Polynomial Regression; ^2^ Gradient Boosting; ^3^ AdaBoost; ^4^ Gaussian naive Bayes; ^5^ Stochastic Gradient Descent; ^6^ Gaussian process; ^7^ Insolation Forest; ^8^ DenseStream; ^9^ Boosting Trees; ^10^ Lasso Regression; ^11^ Decision Support System; ^12^ Generalized Linear Model; ^13^ Linear Regression; ^14^ Gradient Boosted Tree; ^15^ CatBoost, ^16^ classification and regression tree,^17^ Spectral Clustering.

## Data Availability

Not applicable.

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
