# Peer review of "Machine Learning Models and Applications for Early Detection"

_sensors, 2024, doi:10.3390/s24144678_

Round 1

Reviewer 1 Report

Comments and Suggestions for Authors

In this paper, the authors present a systematic literature review on most commonly used machine learning models for early detection, their performance in multiple applications and specifically in fraud detection. The paper is scientifically sound, well written and it is well presented. The review provides useful information on identification of current algorithms and methodologies used in machine learning models for the application of early detection in the referenced areas and in the field of fraud. 

I have a few observations and questions which are listed below: 

  1. My first comment goes with respect to the Article Selection Process. In particular, the choice of “keywords” which then defines the number of articles to be analyzed. Did the authors try other  keywords? For instance, adding “ decision-making” instead of using twice the keyword related to “data analysis”? or any other keyword? This may definitely change the number of articles that will appear.

  2. Do all the articles have the keywords? In many databases this is not the case. Often, the “keywords” are searched in abstracts of the articles. I suggest that the authors clarify this part of the process. 

  3. line 145: When the authors speak about the occurrence of words in selected articles, do they mean in the corpus? in the abstract? This needs to be clarified. 

  4. Figure 1: needs a better resolution, because it can not be read properly. Also I suggest a better description about the figure in the caption. 

  5. line 157: The authors mention 57 articles, but further then only 56 are considered. What happened with an article that dropped out and why? Also it is worth mentioning that 45 articles of multiple areas (line 161) will be described further ahead in section 4. 

  6. It seems that by adding this new keyword “Stacking Ensemble'' (line 156) only 2 more articles have been added (which area?). As you can see the choice of keywords is crucial in defining how many articles will be analyzed and from which area. I suggest that authors comment on this point. 

  7. Table 2: add space between LR and SVM; and between SVM and DT.

  8. My major observation goes to Data characteristics for each area. It is much easier to get better data collection and digitalisation for medicine, than for instance for fraud. This explains the higher number of studies available in the medicine area than in any other. Also, the field of fraud detection presents a lot of confidentiality issues that cannot be resolved so easily. The authors spoke about data balancing in general but I did not see anything mentioning differences and peculiarities of each dataset's acquisitions. Maybe, some other considerations need to be taken into account in order to get a larger scope of papers from other areas than medicine for the analysis of these kinds of models and methodologies.

Author Response

Please see the word document titled: Comments to reviewer 1

Reviewer 2 Report

Comments and Suggestions for Authors

1. It is recommended that the author further expand the scope of the literature review to include top-tier conference and journal papers from the past five years in related fields, ensuring coverage of the latest research progress and achievements.

2. In the review, a more in-depth analysis of the specific application scenarios, advantages, disadvantages, and improvement directions of each machine learning model in early anomaly detection can be provided, offering readers a more comprehensive perspective.

3. While the abstract mentions that MLMs under SBM and SEM implementations achieved accuracies greater than 80% and 90% respectively, it is suggested that the author provide more detailed quantitative comparison results in the main text, including specific experimental setups, dataset descriptions, and evaluation metrics.

4. If possible, it is recommended to include some experimental validation sections to demonstrate the performance differences of different models in early anomaly detection using actual data.

Comments on the Quality of English Language

no

Author Response

Please see the word document titled: Comments to reviewer 2

Reviewer 3 Report

Comments and Suggestions for Authors

The article "Machine Learning Models and Applications for Early Detection" submitted for review is an overview of modern machine learning methods and models that have been widely used in modern scientific applications. However, the article is not without its drawbacks, as follows can be noted: 1) The structure of the article should be revised towards the subject area, which the authors rely on when submitting the article "Fault Diagnosis & Sensors". 2) The authors in the article do not provide a brief description of the models and methods they are investigating. Such a description could improve the quality of the presentation. 3) There are very few illustrations in the work, you can consider adding bar charts that reveal the degree of application of certain methods in different fields. 4) The authors allow large collective references in the review article, for example [1-7]. When it comes to an ordinary article, where the volume of analysis of sources is limited, then collective references can be allowed, but not in such quantity, and in a review article it is important to disclose the characteristics of each source. 5) I think that Figure 2 is a table. It needs to be redone.

Author Response

Please see the word document titled: Comments to reviewer 3

Round 2

Reviewer 2 Report

Comments and Suggestions for Authors

Upon careful re-examination, the author has thoroughly addressed and incorporated all the requirements and suggestions raised in the previous round of review. The manuscript has undergone comprehensive and appropriate revisions. In this round, I have not identified any significant issues or omissions that require further modification. Therefore, I believe that the manuscript has met the publication standards and no additional revision comments are necessary.

Comments on the Quality of English Language

no

Reviewer 3 Report

Comments and Suggestions for Authors

The authors answered the questions about my comments fully and accurately and made a sufficient number of edits to the article material.